# Predictors of Weight-Control Behavior in Healthy Weight and Overweight Korean Middle-Aged Women

**DOI:** 10.3390/ijerph19127546

**Published:** 2022-06-20

**Authors:** Ae-Kyung Chang, Sun-Hui Kim

**Affiliations:** 1College of Nursing Science, Kyung Hee University, Seoul 02447, Korea; akchang@khu.ac.kr; 2Department of Nursing, Graduate School, Kyung Hee University, Seoul 02447, Korea

**Keywords:** body mass index, healthy weight, middle-aged women, weight control, overweight

## Abstract

Although obesity level is considered to influence weight-control behavior, few studies have examined how predictors of weight-control behavior differ according to obesity level. We compared the predictors of weight-control behavior in healthy weight and overweight middle-aged Korean women. This study used a comparative cross-sectional design. In total, 352 middle-aged women (131 overweight and 221 healthy) who visited community centers in eight Korean cities participated in the study. Participants completed self-report questionnaires concerning perceived health, body dissatisfaction, health-related concerns, self-esteem, and weight-control behavior. Scores for weight-control behavior in the overweight group were higher than those in the healthy weight group. Stepwise multiple regression showed that health-related concerns, body dissatisfaction, socioeconomic status, and self-esteem predicted weight-control behavior in the overweight group. Perceived health, socioeconomic status, meal regularity, health-related concerns, and age predicted weight-control behavior in the healthy weight group. The findings indicate that nursing strategies should differ according to obesity level to improve weight-control behavior in middle-aged women. In community or clinical settings, nurses are advised to develop customized weight control programs based on obesity levels in middle-aged women.

## 1. Introduction

Attempting to lose weight is common among people of almost all ages and countries [1]. In particular, women are strongly averse to being overweight for esthetic reasons and tend to be very interested in weight control regardless of their obesity level [2]. Currently, 43.5% of middle-aged Korean women are engaged in weight control [3], whereas 46% of American women follow a weight-reducing diet [4]. However, despite their strong motivation to reduce weight, the obesity rate among Korean women is 35.7%. Specifically, the obesity rate in women aged 40–64 years (42.5%) is three times that of women in their 20s [3]. Female obesity is associated with a greater risk of cancer of the reproductive system, such as breast or ovarian cancer [2], and can trigger chronic diseases, such as hypertension and myocardial infarction; therefore, it leads to serious physical health problems. Furthermore, it reduces the overall quality of life by decreasing self-esteem and inducing psychosocial problems, such as depression or social isolation [2,5]. Accordingly, the World Health Organization has emphasized the need for strategies to help middle-aged women maintain a healthy weight [6].

Treatment methods such as drug therapy, endoscopic treatment, and surgery are effective in managing obesity [7]. However, their benefits cannot be sustained without simultaneous weight control [8]. Weight-control behavior refers to activities practiced for weight reduction or management by healthy individuals [9], and includes a healthy diet, controlled calorie intake, regular exercise, and behavioral modification. Weight-control behavior has been reported to not only reduce blood glucose, serum lipid levels, and blood pressure, but also to decrease depressive symptoms and increase self-esteem [10,11]. Therefore, it is recognized as an important element in obesity management and health-promotion strategies [12]. In South Korea, social interest in weight control has recently increased, and weight control products and programs are widespread. However, according to studies involving Korean women, the most frequently used weight control methods were diet-related and included fasting, meal size reduction, and skipping meals [8,13]; and 30.7% of women engaged in weight-control programs experienced adverse effects [14]. The finding that weight-control behavior can increase, rather than decrease, harmful effects on health suggests a need for an in-depth understanding of weight-control behavior from a health-promotion perspective.

To date, most previous studies involving weight-control behavior have examined interventions [8] and were subject to several limitations; for example, weight loss was either moderate [12] or unsustainable [10,11]. In addition, when similar weight control programs were applied to groups with obesity of different severities, they were ineffective [10]. This finding highlights the need for weight control interventions tailored to individual needs and body mass index (BMI) [15].

The key correlates of weight-control behavior include general characteristics (e.g., age, educational level, marital status, obesity severity, socioeconomic status, and diet), perceived health, body dissatisfaction, self-esteem, and health-related concerns [2,16]. Wang, Liang, and Chen [17] developed a conceptual framework that included body weight, perceived weight status, body dissatisfaction, and weight-control practices. They reported that overweight participants perceived themselves as overweight and experienced body dissatisfaction, which influenced their weight-control behavior. Furthermore, psychological factors have been found to affect weight-control behavior. For example, individuals who judged their current health status as good tended to actively control their weight [18], low self-esteem exerted a negative effect on weight reduction [15], and individuals with strong health-related concerns were likely to engage in healthy weight-control behavior via active health management and sensible eating [19]. Obesity severity has also been shown to directly affect weight-control behavior [2]. Compared with healthy individuals, overweight individuals are less likely to engage in vigorous physical activities or healthy eating. In addition, they tend to report more experiences involving weight control and weight loss failure [20] and often resort to unhealthy weight-control behaviors, such as purging or laxative use [2,11,21]. However, previous research has included obesity severity as a factor influencing weight-control behavior, and few studies have examined how predictors of weight-control behavior differ according to obesity severity [21].

Given the high obesity rate in middle-aged women and the lack of long-term effective obesity treatments, an examination of the differences in factors influencing weight-control behavior between overweight and healthy middle-aged women is required. This could not only aid the development of healthy weight-control programs for middle-aged women but also prevent overweight women from becoming frustrated and choosing unhealthy weight-control methods following weight-reduction failure.

The aims of this study were to examine the effects of general characteristics, body dissatisfaction, perceived health, self-esteem, and health-related concern on weight-control behavior according to obesity severity (overweight or healthy weight) in middle-aged Korean women.

## 2. Materials and Methods

### 2.1. Design

This study used a cross-sectional descriptive design. This study was based on and adapted from a conceptual framework, including body weight, perceived weight status, body dissatisfaction, and weight control practices [17]. According to this model, weight status influences body dissatisfaction, which leads to weight control practices, and psychosocial factors can also influence this process. In addition, we examined perceived health, health-related concerns, and self-esteem, which have influenced weight-control behavior in previous studies (Figure 1) [15,18,19].

### 2.2. Participants

Participants were women aged 35–59 years from eight cities in South Korea. A convenience sample of middle-aged women was recruited for the study. We used the G* Power Program Version 3.1 to estimate the sample size. Assuming a significance level of 0.05, an effect size of 0.13 (medium), and a statistical power of 0.80 in the regression analysis, in accordance with a previous study examining factors influencing weight-control behavior in middle-aged women [2], we estimated that 154 women should be recruited for each group. The inclusion criteria were as follows: aged between 35 and 59 years, ability to understand the study objectives, and ability to understand and complete the survey. The exclusion criterion was physical disability that could limit weight-control behavior (e.g., paralysis, fracture, or amputation).

### 2.3. Data Collection

#### 2.3.1. General Characteristics

General characteristics included age, BMI, alcohol consumption, smoking status, marital status, employment status, educational level, meal regularity, socioeconomic status, and disease status. To compute BMI, we used participants’ self-reported height and weight. BMI was calculated by dividing the weight (kg) by the height squared (m^2^). Standard BMI cut-off points for the Korean population [22] were used to categorize weight status as follows: underweight: <18.5 kg/m^2^, healthy weight: 18.5–22.9 kg/m^2^, overweight: 23–24.9 kg/m^2^, and obese: ≥25 kg/m^2^. No participants were classified as underweight; therefore, participants were of healthy weight, overweight, or obese. In all analyses, women with BMIs < 23 kg/m^2^ and ≥23 kg/m^2^ were considered to be of healthy weight and overweight, respectively.

#### 2.3.2. Perceived Health

Perceived health was measured using the question: “How would you assess your current health status?” Responses ranged from one (very unhealthy) to five (very healthy).

#### 2.3.3. Body Dissatisfaction

Body dissatisfaction was measured using the body dissatisfaction subscale from the Multidimensional Eating Disorder Inventory developed by Garner, Olmstead, and Polivy [23]. The scale consists of 19 items measuring the levels of dissatisfaction with the upper, middle, and lower parts of the body. Responses were provided using a five-point scale ranging from one to five, with higher scores indicating greater dissatisfaction with the body. Cronbach’s α for the scale was 0.91 in Garner et al.’s study [23] and 0.84 in the current study.

#### 2.3.4. Health-Related Concerns

Health-related concerns were measured using the Health-Related Concern Scale developed by Park et al. [24] to evaluate the extent to which individuals were concerned about their health. The scale consists of five items that measure interest in health-related knowledge, activities, and attitudes. Responses were provided on a five-point scale, with higher scores indicating stronger health-related concerns. Cronbach’s α for the scale was 0.74 in Park et al.’s study [24] and 0.76 in the current study.

#### 2.3.5. Self-Esteem

Self-esteem was measured using the Rosenberg Self-Esteem Scale [25]. The scale consists of 10 items, with responses provided on a four-point scale, with higher scores indicating higher levels of self-esteem. Cronbach’s α was 0.81 in Rosenberg’s study [25] and 0.78 in the current study.

#### 2.3.6. Weight-Control Behavior

Weight-control behavior was measured using the Weight-Control Practice Tool developed by Jung in 1989 [9]. The scale consists of 15 items that measure the use of diet, exercise, behavioral modification, and drug therapy to lose or maintain weight. Responses were provided using a five-point scale, with higher scores indicating higher levels of weight-control behavior. Cronbach’s α for the scale was 0.76 in Jung’s study [9] and 0.78 in the current study.

### 2.4. Data Collection

Data were collected in September and October 2020. The researchers visited community centers in eight small- and medium-sized cities and requested relevant officials’ cooperation with data collection. Participants were recruited via messages displayed at community centers. Women who expressed an interest in the research were contacted by the researchers in person or via telephone and provided detailed information regarding the study; a date was arranged for data collection. Of these, 358 women met the inclusion criteria. Prior to data collection, eight nursing students who served as research assistants attended a 1 h training session to learn about the study objectives, data collection method, and communication skills in interactions with participants. We examined the inter-rater reliability (Fleiss Kappa = 0.89) using pre-tests involving the survey. The participants received a small gift for their participation.

### 2.5. Ethical Considerations

Ethical approval for the study was granted by the institutional review board of Kyung Hee University (approval no. KHSIRB-20-006). All participants provided written informed consent to participate in the study.

### 2.6. Data Analysis

The data were analyzed using SPSS version 19.0 (IBM Corp., Armonk, NY, USA), as follows: initial *t*-tests and chi-square tests were performed to examine differences in general characteristics, psychosocial factors, and weight-control behavior between the overweight and healthy weight groups. Further *t*-tests and analyses of variance were performed to examine the differences in factors affecting weight-control behavior between the two groups. A post-hoc analysis was performed using Scheffé’s test. Pearson’s correlation coefficients were calculated to examine the relationships between independent variables and weight-control behavior in each group. Stepwise multiple regression analysis was then performed to identify the predictors of weight-control behavior in both groups. The significance level was set at 0.05.

## 3. Results

### 3.1. Differences in Main Variables and Weight-Control Behavior

Initially, 365 women expressed interest in the study; of these, 358 met the inclusion criteria. In total, 358 surveys were returned, of which six were excluded because of incomplete responses. Thus, finally, responses from 352 participants were analyzed. Of the 352 participants, 131 (37.2%) and 221 (62.8%) were classified as overweight and healthy, respectively, according to BMI values. The comparison of general characteristics between the groups showed that, relative to the healthy weight group, the overweight group was older (mean age = 45.12 vs. 47.05 years; *p* = 0.026) and had higher mean BMI values (20.49 vs. 26.26; *p* < 0.001).

The proportion of women in the overweight group who reported experiencing disease was 37.4%, which was higher than that observed in the healthy weight group (21.3%; *p* < 0.001). The perceived health score in the overweight group (3.07 out of 5) was significantly lower than that observed in the healthy weight group (3.28 out of 5, *p* = 0.017). The comparison of psychosocial characteristics between the groups showed that body dissatisfaction was significantly higher in the overweight group than in the healthy weight group (*p* = 0.005); however, health-related concerns and self-esteem did not differ significantly between the groups. In addition, the weight-control behavior score in the overweight group was significantly higher than that in the healthy weight group (*p* < 0.001; Table 1).

### 3.2. Comparison of Factors Affecting Weight-Control Behavior

There were significant differences in weight-control behavior according to disease and socioeconomic status in the overweight group (Table 2). Specifically, weight-control behavior scores in participants with disease were higher than those observed in participants without disease (*p* = 0.032).

Weight-control behavior differed significantly according to marital status, meal regularity, and socioeconomic status in the healthy weight group. Married participants and those who ate regular meals exhibited higher scores for weight-control behavior relative to those observed for other participants (*p* < 0.001). However, scores in participants with middle or high socioeconomic status were higher than those observed for participants with low socioeconomic status (*p* = 0.001).

### 3.3. Comparison of Correlations between Weight-Control Behavior and Related Factors

Weight-control behavior was positively correlated with body dissatisfaction (*r* = 0.42, *p* < 0.001), self-esteem (*r* = 0.41, *p* < 0.001), and health-related concerns (*r* = 0.48, *p* < 0.001) in the overweight group. In contrast, weight-control behavior was positively correlated with age (*r* = 0.23, *p* < 0.001), perceived health (*r* = 0.42, *p* < 0.001), self-esteem (*r* = 0.22, *p* = 0.001), and health-related concerns (*r* = 0.24, *p* < 0.001) in the healthy weight group.

### 3.4. Comparison of Predictors of Weight-Control Behavior

The results of the stepwise multiple regression analysis were as follows. The predictive model for weight-control behavior in the overweight group was significant (*F* = 27.81, *p* < 0.001), and health-related concerns, body dissatisfaction, socioeconomic status, and self-esteem were identified as significant predictors and explained 46.9% of the variance in weight-control behavior (Table 3). Of these variables, health-related concerns explained the highest proportion (23.2%) of the variance in weight-control behavior, followed by body dissatisfaction (16.0%), and socioeconomic status (5.0%). With respect to the model’s goodness of fit, tolerance limits exceeded 0.10 (0.73 to 0.96), and variance inflation factors were below 10 (1.00–1.36), indicating the absence of multicollinearity (Table 4).

The predictive model for weight-control behavior in the healthy weight group was also significant (*F* = 21.81, *p* < 0.001). In this model, perceived health, socioeconomic status, meal regularity, health-related concerns, and age were identified as significant predictors and explained 33.7% of the variance in weight-control behavior. Perceived health explained the highest proportion (17.7%) of the variance in weight-control behavior, followed by socioeconomic status (5.8%) and meal regularity (4.8%). With respect to the model’s goodness of fit, tolerance limits exceeded 0.10 (0.92 to 0.97), and variance inflation factors were below 10 (1.02–1.08), indicating the absence of multicollinearity (Table 4).

## 4. Discussion

This study examined factors affecting weight-control behavior according to obesity severity in middle-aged Korean women. The results showed that weight-control behavior scores in the overweight group were higher than those observed in the healthy weight group. This finding is consistent with those of previous studies indicating that obese women exerted greater effort to reduce their weight than non-overweight women [16], and weight-control behavior scores in overweight middle-aged Korean women were higher than those in their underweight counterparts [12]. In modern society, a slim body type is considered ideal for women; therefore, obese women are pressured to lose weight. The results indicated that as obesity increased in middle-aged women, they begin to show greater engagement in weight-control behavior. However, overweight women often fail to reduce their weight through diet or exercise [18]. In addition, as the difference between their actual and ideal weight increases, they tend to seek a rapid solution and engage in unhealthy practices (e.g., vomiting and use of diet pills) to lose weight [5]. A previous study indicated that overweight girls were twice as likely to engage in unhealthy weight-control behaviors than healthy girls [21]. Therefore, interventions should be developed to screen overweight women for harmful weight management methods and to promote healthy lifestyle habits.

The results showed that socioeconomic status and health-related concerns predicted weight-control behavior in both the overweight and healthy weight groups. Body dissatisfaction and self-esteem were identified as predictors only in the overweight group, whereas perceived health, meal regularity, and age were identified as predictors only in the healthy weight group. These findings support the results of a previous study [18], in which Korean women with higher socioeconomic status and stronger health-related concerns were more likely to use weight reduction methods relative to those with lower socioeconomic status and weaker health-related concerns, regardless of obesity severity. In addition, body dissatisfaction and self-esteem predicted weight control practices in overweight women [10], whereas perceived health and health-related concerns were predictors of weight-control behavior in women of a healthy weight [5].

As a common predictor of weight-control behavior in both groups, socioeconomic status was a significant predictor of weight-control behavior in middle-aged women. This finding is consistent with those of a previous study [26], in which individuals experienced a financial burden when they engaged in weight-control behavior, and those with high socioeconomic status were 1.23 times more likely to practice weight-control behavior than those with low socioeconomic status. According to a recent Korean study, lower socioeconomic status limits individuals’ access to high-quality foods and healthy activities, including weight control programs [27]. Therefore, consideration of the increased social cost of obesity should include the establishment of national strategies to improve access to exercise facilities and obesity management programs for individuals of low socioeconomic status. In addition, the finding that health-related concerns exerted a significant effect on middle-aged women’s weight-control behavior supports the findings of previous studies [5] in which women in their 20s engaged in weight-control behavior for esthetic reasons, whereas those in their 40s and 50s engaged in such behavior for health reasons. This finding is also consistent with a study in which middle-aged women became more motivated to control their weight as they aged and health problems occurred in their spouses or parents [19]. Considering that individuals with strong health-related concerns tend to manage their health actively and follow a sensible diet [18], women with weak health-related concerns should be screened to promote healthy behaviors and encourage participation in health education programs. This could lead to healthy lifestyle habits and, ultimately, exert a positive influence on weight control.

In addition to the influential factors common to both the overweight and healthy weight groups, body dissatisfaction and self-esteem only affected the weight-control behavior of the overweight group. Previous studies have identified body dissatisfaction as a main predictor of weight-control behavior in obese or overweight women [5,28]; this finding is consistent with the current study. Furthermore, body dissatisfaction has a favorable effect on weight control in overweight or borderline overweight individuals, as it could induce healthy weight-control behavior, such as increasing fruit and vegetable intake or engaging in regular physical exercise [17]. However, in a study involving teenagers with a healthy weight, body dissatisfaction led to imprudent weight-control activities that resulted in eating disorders [29]. Therefore, weight control programs should manage individuals’ body dissatisfaction according to the severity of obesity.

The finding that self-esteem predicted weight-control behavior in the overweight group is consistent with the results of previous studies in which self-esteem was the most powerful predictor in overweight and obese middle-aged women [15,30]. Teixeria et al. [10] conducted a review study involving obese women of all ages and concluded that increasing self-esteem is an effective method for maintaining weight loss, because increased self-esteem decreases negative body image, which helps overweight individuals to maintain weight-control behavior. Accordingly, there is a need for customized weight management programs for overweight women with low self-esteem, which should include intervention strategies to help them reclaim positive attitudes towards their body image and increase their self-esteem.

Perceived health, meal regularity, and age affected weight-control behavior only in the healthy weight group. The finding that perceived health predicted weight-control behavior is supported by previous studies in which healthy Korean women with good perceived health were 1.41 times more likely to engage in weight-control behavior than those with poor perceived health among Korean women with healthy weight [2]. This indicates that the perception of one’s health as good could also increase the likelihood of practicing weight-control behavior. Additionally, meal regularity was a significant predictor of weight-control behavior in the healthy weight group. This finding is similar to those of previous studies in which individuals of a healthy weight exhibited healthier eating habits, such as eating regular meals, relative to those of overweight individuals [31], and people who ate breakfast regularly were more likely to engage in weight-control behavior than those who did not eat breakfast regularly [12]. The reduction of calorie intake has generally been recommended as a means of reducing weight; however, several studies have shown that irregular eating patterns or skipping meals predicted greater weight increases, and the researchers concluded that limiting meal frequency could be ineffective as a weight-maintenance method [12,32]. Therefore, based on the current finding that meal regularity affected weight-control behavior in the healthy weight group, experimental research should be conducted to determine whether eating regular meals without limiting food intake can be effective in promoting weight control and maintenance in healthy women. In addition, the finding that meal regularity was not an effective method of weight control for overweight women supports the conclusion that weight management strategies should be tailored to individual characteristics (e.g., obesity severity). Furthermore, age was a significant predictor of weight-control behavior, which is consistent with previous findings indicating that age exerts a direct effect on weight-control behavior in healthy Korean women, and levels of weight-control behavior were highest in women older than 40 years of age [2]. Women’s weight generally begins to increase after birth, and they are likely to become overweight as they age. According to a qualitative study examining weight control experiences in middle-aged Korean women [19], most participants engage in weight-control behavior as they age. Therefore, it is necessary to develop weight-control programs that meet the needs of middle-aged women, can be followed easily, are enjoyable, and provide ongoing incentives to continue participation.

This study had some limitations. For example, it utilized a cross-sectional rather than a longitudinal study design; therefore, we could not infer causality in the relationships between the study variables, implying the need for a replication study. Additionally, because the study was conducted in South Korea, the results cannot be generalized to other countries or cultural backgrounds.

## 5. Conclusions

The weight-control behavior scores were higher in the overweight group than in the healthy weight group. Socioeconomic status and health-related concerns predicted weight-control behavior in both overweight and healthy weight groups. Body dissatisfaction and self-esteem were identified as predictors only in the overweight group, whereas perceived health, meal regularity, and age were identified as predictors in the healthy weight group.

In community or clinical settings, nurses are advised to develop customized weight control programs based on obesity levels in middle-aged women. In particular, nurses might play an important role in screening for overweight women with lower socioeconomic status, body dissatisfaction, poor self-esteem, and higher health-related concerns to promote healthy weight-control behavior. In addition, when performing regular health check-ups and healthy lifestyle education for healthy women, nurses should focus on these women’s health-related concerns and meal regularity to help them maintain weight.

## Figures and Tables

**Figure 1 ijerph-19-07546-f001:**
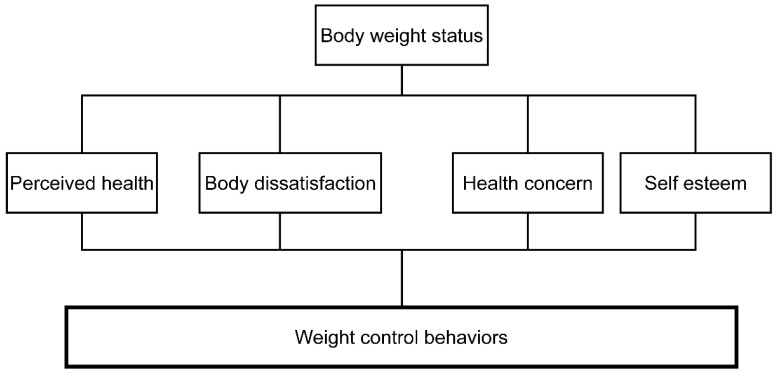
Conceptual framework.

**Table 1 ijerph-19-07546-t001:** Differences in main variables and weight-control behavior between the overweight and healthy weight groups (*n* = 352).

Characteristics	Categories	Overweight (*n* = 131) M ± SD or *n* (%)	Healthy Weight (*n* = 221) M ± SD or *n* (%)	χ^2^ or *t*	*p*
Age (years)		47.05 ± 7.60	45.12 ± 7.89	2.23	0.026
BMI (kg/m^2^)		26.26 ± 2.24	20.49 ± 1.56	21.44	<0.001
Alcohol consumption	Yes	56 (42.7)	80 (36.2)	1.48	0.134
	No	75 (57.3)	141 (63.8)		
Smoking	Yes	4 (3.1)	3 (1.4)	1.21	0.431
	No	127 (96.9)	218 (98.6)		
Spouse	Yes	113 (86.3)	185 (87.3)	0.41	0.545
	No	18 (13.7)	36 (16.3)		
Employment status	Yes	90 (69.2)	147 (66.5)	0.27	0.638
	No	40 (30.8)	74 (33.5)		
Education	High school degree or less	14 (10.7)	22 (10.0)	0.14	0.857
	College graduate or more	117 (89.3)	199 (90.0)		
Meal regularity	Regular	84 (64.1)	146 (66.1)	0.13	0.729
	Irregular	47 (35.9)	75(33.9)		
Disease	Yes	49 (37.4)	47 (21.3)	10.79	0.001
	No	82 (62.6)	174 (78.7)		
Socioeconomic status	High	6 (4.6)	11 (5.0)	2.38	0.304
	Middle	97 (74.0)	177 (80.1)		
	Low	28 (21.4)	33 (14.9)		
Perceived health		3.07 ± 0.80	3.28 ± 0.76	−2.40	0.017
Body dissatisfaction		56.54 ± 6.64	54.36 ± 7.09	2.85	0.005
Self-esteem		28.64 ± 3.84	28.69 ± 3.48	−0.12	0.905
Health-related concern		18.95 ± 2.92	19.01 ± 3.69	−0.15	0.875
Weight-control behavior		49.52 ± 7.39	46.18 ± 7.24	4.14	<0.001

M ± SD, mean ± standard deviation; BMI, body mass index.

**Table 2 ijerph-19-07546-t002:** Comparison of factors affecting weight-control behavior in the overweight and healthy weight groups (*n* = 352).

Characteristics	Categories	Overweight (*n* = 131)		Healthy Weight (*n* = 221)	
		M ± SD	*t* or F/*p* (Post Hoc)	M ± SD or *n* (%)	χ^2^ or F/*p* (Post Hoc)
Alcohol consumption	Yes	48.76 ± 8.37	−1.09/0.278	44.68 ± 7.26	−2.06/0.051
	No	49.96 ± 6.76		47.29 ± 7.06	
Smoking	Yes	42.33 ± 10.46	−0.87/0.475	35.50 ± 9.46	−2.73/0.069
	No	49.62 ± 7.26		44.57 ± 0.6.83	
Spouse	Yes	49.18 ± 7.06	0.06/0.996	47.21 ± 7.44	3.19/0.002
	No	49.17 ± 8.51		43.00 ± 6.06	
Employment status	Yes	49.18 ± 7.17	−0.95/0.339	46.33 ± 6.99	−0.33/0.741
	No	50.22 ± 7.81		45.88 ± 7.91	
Education	High school degree or less	51.82 ± 6.44	1.53/0.125	48.36 ± 6.46	1.19/0.234
	College graduate or more	49.27 ± 7.45		45.91 ± 7.30	
Meal regularity	Regular	50.49 ± 7.33	0.65/0.511	48.09 ± 7.90	4.58/<0.001
	Irregular	49.62 ± 7.10		43.48 ± 6.72	
Disease	Yes	48.43 ± 8.00	−2.16/0.032	46.45 ± 7.84	0.08/0.935
	No	51.22 ± 6.57		46.55 ± 7.28	
Perceived economic status	High ^a^	53.73 ± 5.21	12.86/<0.001a > b	50.00 ± 5.83	7.30/0.001a > b
	Middle ^a^	50.28 ± 7.20		47.18 ± 6.46	
	Low ^b^	44.09 ± 6.48		41.89 ± 8.46	

M ± SD, mean ± standard deviation. ^a^: Perceived economic status: middle or high; ^b^: perceived economic status: low.

**Table 3 ijerph-19-07546-t003:** Comparison of correlations between weight-control behavior and related factors in the overweight and healthy weight groups (*n* = 352).

Variables	Overweight (*n* = 131)		Healthy-Weight (*n* = 221)	
	*r*	*p*	*r*	*p*
Age	0.16	0.053	0.23	<0.001
Perceived health	0.06	0.469	0.42	<0.001
Body dissatisfaction	0.42	<0.001	0.02	0.757
Self-esteem	0.41	<0.001	0.22	0.001
Health-related concern	0.48	<0.001	0.24	<0.001

**Table 4 ijerph-19-07546-t004:** Comparison of predictors of weight-control behavior in the overweight and healthy weight groups (*n* = 352).

Variables	B	S.E	β	*t*	*p*	Partial R^2^	R^2^	Adjusted R^2^	F/*p*
Overweight group									
Constant	−1.725	4.848		0.710	0.723		0.469	0.452	27.81/<0.001
Health-related concern	0.844	0.187	0.341	5.516	<0.001	0.232			
Body dissatisfaction	0.348	0.059	0.384	5.899	<0.001	0.161			
Socioeconomic status	3.786	1.161	0.215	3.262	0.001	0.052			
Self-esteem	0.338	0.143	0.180	2.368	0.019	0.024			
Healthy weight group									
Constant	20.910	3.372		6.200	<0.001		0.337	0.321	21.81/<0.001
Perceived health	3.110	0.555	0.324	5.601	<0.001	0.177			
Socioeconomic status	4.196	1.183	0.203	3.545	<0.001	0.058			
Meal regularity	3.169	0.881	0.203	3.595	<0.001	0.048			
Health-related concern	0.369	0.113	0.185	3.277	0.001	0.036			
Age	0.127	0.053	0.136	2.388	0.018	0.018			

## Data Availability

Not applicable.

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
