# Peer review of "Predictors of Weight-Control Behavior in Healthy Weight and Overweight Korean Middle-Aged Women"

_ijerph, 2022, doi:10.3390/ijerph19127546_

Round 1

Reviewer 1 Report

I would like to thank to the editor the opportunity of reviewing this interesting study aimed to examine the differences in factors influencing weight-control behavior between overweight and healthy middle-aged women is required. The manuscript is original due to its theme and of great interest to the readers of the International Journal of Environmental Research and Public Health. Overall it seems that a lot of work has been done already, and Authors have used relevant tools. However, authors must clarify/add information about some issues what I indicate below:

Introduction

- The introduction is well written and contextualize adequately the study. However, at the end of the introduction, part of the results obtained in the work "Thus, we Taimed examined the effects of general characteristics, body dissatisfaction, perceived health, self-esteem, and health-related concerns on weight-control behavior according to obesity severity (overweight or healthy weight) in middle-aged Korean women. We found that weight-control behavior scores were higher in the overweight group than in the healthy weight group. Socioeconomic status and health-related concerns predicted weight-control behavior in both overweight and healthy weight groups. Bodysuit dissatisfaction and self-esteem were identified as predictors only in the overweight group, whereas perceived health, meal regularity, and age were identified as predictors in the healthy weight group"

Method

- The BMI values used belong to those published in 2002 by the WHO, however, the latest values provided by the WHO in 2021 refer to overweight is a "BMI greater than or equal to 25; and obesity is a BMI greater than or equal to 30".

https://www.who.int/news-room/fact-sheets/detail/obesity-and-overweight

Author Response

Comment 1

The introduction is well written and contextualize adequately the study. However, at the end of the introduction, part of the results obtained in the work "Thus, we aimed examined the effects of general characteristics, body dissatisfaction, perceived health, self-esteem, and health-related concerns on weight-control behavior according to obesity severity (overweight or healthy weight) in middle-aged Korean women. We found that weight-control behavior scores were higher in the overweight group than in the healthy weight group. Socioeconomic status and health-related concerns predicted weight-control behavior in both overweight and healthy weight groups. Bodysuit dissatisfaction and self-esteem were identified as predictors only in the overweight group, whereas perceived health, meal regularity, and age were identified as predictors in the healthy weight group"

  • Thank you for pointing this out. We agree with this comment. Therefore, we have deleted the results in the introduction part.

Comment 2

The BMI values used belong to those published in 2002 by the WHO, however, the latest values provided by the WHO in 2021 refer to overweight is a "BMI greater than or equal to 25; and obesity is a BMI greater than or equal to 30".

https://www.who.int/news-room/fact-sheets/detail/obesity-and-overweight

  • Thank you for pointing this out. However, we believe that World Health Organization's Asia-Pacific region and the Korean Journal of Obesity use this standard that we have described.

Reviewer 2 Report

This is a comparative cross-sectional design study that aims to examine the effects of several predictors on weight-control behavior according to obesity severity in middle-aged Korean women.

Title: Indicate cross-sectional design with a commonly used term. Given the specific cultural context of Korean women, consider including that in the title too. 

Introduction: Well written. The information about South Korea should be a separate paragraph. In the last paragraph from “We found that…” and onwards, please remove and reserve this information for the Results section. Instead, can state prespecified hypotheses.

Methods: 2.1 - If possible, note other studies that used this conceptual framework for similar purposes. 2.2 - Info about how many surveys were returned, etc. should be reserved for Results. 2.4 - Since this is essentially your procedures section please detail how confidentiality was maintained and how informed consent was ensured. The sentence about how many women expressed interest and were included should be removed and reserved for Results.

Results: Ensure the information about the number of individuals at each stage of the study (ie, potentially eligible, included, etc.) is reported here. Consider using a flow diagram. Address if there are any exposures/potential confounders and describe results of the general characteristics (alcohol, smoking, etc) in 3.1. 

Discussion: Second last paragraph ought to be separated by the various factors mentioned for better clarity. The interpretation is well written and may be improved by making paragraphs more concise. In the last paragraph, the statement about self-report is confusing as self-report usually means self-administered survey. The Data collection section seemed to convey that Research Assistant nurses administered the survey which leads to different biases and limitations - please clarify and address this.

Conclusion: The emphasis on a nurse’s role makes sense but is newly introduced in the conclusion. Please elaborate and stress this somewhere in the Discussion if you want to maintain it as a key point in this section. May be better to extend this point to other healthcare professionals too - family physicians, nutritionists, etc.

Writing style: Please review all sections for spelling, grammar, and typos. Paragraph structure is well done but can be enhanced with clear topic sentences and concluding sentences. 

Author Response

Comments and Suggestions for Authors

This is a comparative cross-sectional design study that aims to examine the effects of several predictors on weight-control behavior according to obesity severity in middle-aged Korean women.

  1. Title: Indicate cross-sectional design with a commonly used term. Given the specific cultural context of Korean women, consider including that in the title too.
  • Thank you for pointing this out. Accordingly, Korean was added to the title in consideration of the specific cultural context of Korean women.

  1. Introduction: Well written. The information about South Korea should be a separate paragraph. In the last paragraph from “We found that…” and onwards, please remove and reserve this information for the Results section. Instead, can state prespecified hypotheses.
  • Thank you for pointing this out. Accordingly, we have deleted the results in the introduction part.

  1. Methods: 2.1 - If possible, note other studies that used this conceptual framework for similar purposes.
  • Thank you for pointing this out. However, currently, it is designed based on that framework, and the dependent variable has been determined; therefore, it cannot be changed now.

  1. Info about how many surveys were returned, etc. should be reserved for Results Include information on how many surveys were returned, etc. in the results

- Thank you for pointing this out. Accordingly, we deleted the sentence regarding the actual number of participants from here and included it along with the other numbers in the results section.

  1. Since this is essentially your procedures section please detail how confidentiality was maintained and how informed consent was ensured. The sentence about how many women expressed interest and were included should be removed and reserved for Results.
  • Thank you for pointing this out. Accordingly, Information on how many women expressed interest and were included was deleted and added to the results section. Initially, 365 women expressed interest in the study; of these, 358 met the inclusion criteria. In total, 358 surveys were returned, of which six were excluded because of incomplete responses. Finally, responses from 352 participants were analyzed.

  1. Results: Ensure the information about the number of individuals at each stage of the study (ie, potentially eligible, included, etc.) is reported here. Consider using a flow diagram. Address if there are any exposures/potential confounders and describe results of the general characteristics (alcohol, smoking, etc) in 3.1
  • Thank you for pointing this out. General characteristics were described, including age, BMI, alcohol consumption, smoking status, marital status, employment status, education level, diet regularity, socioeconomic status, and disease status.

  1. Discussion: Second last paragraph ought to be separated by the various factors mentioned for better clarity. The interpretation is well written and may be improved by making paragraphs more concise. In the last paragraph, the statement about self-report is confusing as self-report usually means self-administered survey. The Data collection section seemed to convey that Research Assistant nurses administered the survey which leads to different biases and limitations - please clarify and address this.
  • Thank you for pointing this out. Accordingly, we deleted the last paragraph because we thought the meaning of self-report was ambiguous. In the data collection section, the inter-rater reliability (Fleiss Kappa=0.89) was added in the part where the research assistant conducted the survey.

  1. Conclusion: The emphasis on a nurse’s role makes sense but is newly introduced in the conclusion. Please elaborate and stress this somewhere in the Discussion if you want to maintain it as a key point in this section. May be better to extend this point to other healthcare professionals too - family physicians, nutritionists, etc.
  • Thank you for pointing this out. Accordingly, we added that we would like to extend this to other health care professionals such as family medicine specialists and nutritionists.
  1. Writing style: Please review all sections for spelling, grammar, and typos. Paragraph structure is well done but can be enhanced with clear topic sentences and concluding sentences.

Thank you for pointing this out. Accordingly, we have checked the entire document for typos and spellings.

Reviewer 3 Report

Introduction:

What is the  purpose of this study?

Thus, we Taimed examined the effects of general characteristics, body dissatisfaction, perceived health, self-esteem, and health-related concerns on weight-control behavior according to obesity severity (overweight or healthy weight) in middle-aged Korean women (Grammar at the start of this is not correct).

Author Response

Comments and Suggestions for Authors

Introduction:

What is the purpose of this study?

Thus, we aimed examined the effects of general characteristics, body dissatisfaction, perceived health, self-esteem, and health-related concerns on weight-control behavior according to obesity severity (overweight or healthy weight) in middle-aged Korean women (Grammar at the start of this is not correct).

  • Thank you for pointing this out. Accordingly, we have checked the entire document for typos and spellings and changed the sentence to “The aims of this study were to examine the effects of general characteristics, body dissatisfaction, perceived health, self-esteem, and health-related concern on weight-control behaviour according to obesity severity (overweight or healthy weight) in middle-aged Korean women.”
